# Baculoviruses manipulate host lipid metabolism via adipokinetic hormone signaling to induce climbing behavior

**Lin Zhu** [1], **Yuqing Xie**[1], **Chenxi Liu**[1], **Jie Cheng**[2], **Zhongjian Shen**[3], **Xiaoming Liu**[4], **Limei Cai**[1], **Xinyuan Ning**[1], **Songdou Zhang**[1], **Zhen Li**[1], **Qiuying Huang**[5], **Xiaoxia Liu** [1]*

**1** Department of Entomology and MOA Key Lab of Pest Monitoring and Green Management, College of Plant Protection, China Agricultural University, Beijing, China, **2** Department of Entomology College of Plant Protection, Shanxi Agricultural University, Jinzhong, China, **3** State Key Laboratory for Biology of Plant Diseases and Insect Pests, Key Laboratory of Natural Enemy Insects of Ministry of Agriculture and Rural Affairs, Institute of Plant Protection, Chinese Academy of Agricultural Sciences, Beijing, China, **4** State Key Laboratory of Wheat and Maize Crop Science, Henan International Laboratory for Green Pest Control, College of Plant Protection, Henan Agricultural University, Zhengzhou, China, **5** Hubei Insect Resources Utilization and Sustainable Pest Management Key Laboratory, Huazhong Agricultural University, Wuhan, China

\* liuxiaoxia611@cau.edu.cn

**Data Availability Statement:** All the gene sequences in this study were from genomic

## Abstract

Baculoviruses can induce climbing behavior in caterpillar hosts, which provides an excellent model for studying parasite manipulation of host behavior. Herein, we found that *Helicoverpa armigera* single nucleopolyhedrovirus (HearNPV) promoted lipid metabolism of infected *H. armigera* larvae, and changes in lipid metabolism can affect climbing behavior. Therefore, understanding the molecular mechanisms between lipid metabolism and climbing behavior is particularly important. In this study, we found adipokinetic hormone 1 (*HaAKH1*), adipokinetic hormone 2 (*HaAKH2*) and their receptor *HaAKHR* were essential for promoting lipid metabolism and climbing behavior in response to HearNPV infection. Both molecular docking result and $Ca^{2+}$ imaging showed that both HaAKH1 and HaAKH2 could interact with HaAKHR. Knockdown of *HaAKH1*, *HaAKH2* and *HaAKHR* resulted in not only the accumulation of triacylglycerol (TAG), but also the reduction of the replication of HearNPV and the crawling ability of infected *H. armigera* larvae, resulting in a decrease in the final death height of the infected larvae. We further validated this conclusion by injecting active peptides of HaAKH1 and HaAKH2 to infected larvae. In addition, we investigated the downstream of HaAKH signaling and found that hormone-sensitive lipase (*HaHSL*) changed with changes in *HaAKH* signaling and *HaHSL* played the same role as HaAKH signaling. These findings not only revealed the mechanism by which parasites manipulated host lipid metabolism, but more significantly, explored the relationship between lipid metabolism and behavioral changes of hosts manipulated by parasites, broadening our understanding of the phenomenon of parasites manipulating host behavioral changes.

sequence of Helicoverpa armigera (GenBank Accession: GCA_030705265.1).

**Funding:** This work was supported by the National Key R&D Program of China (grant number 2023YFD1400700 to QH) and the National Natural Science Foundation of China (grant number 32202293 to XL). The funders had no role in study design, data collection and analysis, decision to publish, or preparation of the manuscript.

**Competing interests:** The authors have declared that no competing interests exist.

## Author summary

The phenomenon of parasites manipulating changes in host behavior is widely present in nature. During the long-term coevolution of parasites and hosts, parasites have evolved adaptive strategies to regulate the physiological and behavioral changes of their hosts, in order to facilitate their own reproduction and spread. Here, we find that baculoviruses manipulate the energy metabolism of Lepidoptera larvae to facilitate their own replication and promote host locomotion, inducing the climbing behavior of infected hosts, referred to as "tree-top disease". Specifically, we find that the baculovirus HearNPV promotes lipid metabolism by inducing the expression of *Helicoverpa armigera* adipokinetic hormone HaAKH1 and HaAKH2, activating the receptor HaAKHR, and promoting the expression of the lipase HaHSL, providing energy for HearNPV replication and host locomotion, thereby achieving parasite manipulation of host lipid homeostasis. Our research provides innovative insights into the mechanisms by which parasites manipulate host lipid homeostasis and behavioral changes, as well as the interactions between parasites and hosts.

## Introduction

Parasites often manipulate their hosts, altering their behavior, appearance, or physiology to promote their own survival and transmission [1,2]. A well-known example is the manipulation of behavior in Lepidoptera caterpillars infected with baculoviruses [3], which induce hyperactivity and climbing behavior, ultimately leading to the caterpillars' death and liquefaction at the tops of plants, referred to as "tree-top disease" [4,5]. Despite the prevalence of host manipulation by parasites [2], the underlying mechanisms remain poorly understood [6].

Parasite development is closely tied to host metabolism, with coevolution prompting parasites to manipulate host physiology to meet their specific nutritional needs [7–11]. Insects, for instance, rely on lipids as key energy sources during various life stages, with triacylglycerols (TAGs) stored in the fat body as the primary lipid reserve [12,13]. An endoparasitic wasp *Cotesia vestalis* of *Plutella xylostella* larvae uses symbiotic bracovirus as a weapon to manipulate host lipid levels, leading to a nutritional lipid level suitable for the development of *C. vestalis* wasps [14]. However, research on baculovirus-induced changes in host lipid metabolism, and its link to behavioral manipulation, remains limited.

Adipokinetic hormone (AKH), a neuropeptide that regulates lipid metabolism, plays a critical role in managing energy reserves in various insects [15]. AKH is synthesized in the corpora cardiaca and binds to the AKHR receptor in the fat body [16–20]. Recent studies have highlighted the role of AKH signaling in host-parasite interactions. For instance, injecting AKH into locusts infected with *Metarhizium anisopliae* accelerates their mortality [21], and the application of the entomopathogenic fungus *Isaria fumosorosea* increases AKH levels in *Periplaneta americana* [22], enhancing the fungus's lethality.

AKH triggers the breakdown of TAGs in the fat body by activating lipases through the cAMP/PKA signaling pathway [23]. This process involves lipid storage droplet proteins and triglyceride lipases, including Brummer (Bmm) and hormone-sensitive lipase (HSL) [23,24]. While AKH signaling has been implicated in lipid metabolism, its role in behavioral changes induced by baculoviruses in *Helicoverpa armigera* has not been extensively studied.

This study reveals that HearNPV hijacks the AKH signaling pathway to enhance lipid metabolism and promote climbing behavior in *H. armigera*. Our findings contribute to a deeper understanding of how parasites manipulate host behavior at the molecular level and open new possibilities for enhancing baculovirus-based pest control strategies.

## Materials and methods

### Insect rearing and baculovirus infection

The colony of *H. armigera* was established from a natural population in Zhengzhou, Henan Province, and was subsequently reared in the laboratory under standardized conditions of $26 \pm 1°C$, $70 \pm 10\%$ RH, and a 14:10 (L:D) photoperiod. Larvae were reared with artificial diet in glass test tubes (85 mm × 22 mm diameter) plugged with cotton [25]. The pupae were transferred to a plastic frame cage (40 × 20 × 20 cm) covered with gauze. Emerging adult moths were fed 10% honey solution and laid eggs on the gauze.

The HearNPV used in this study was obtained from Henan Jiyuan Baiyun Industry as a purified freeze-dried powder ($5 \times 10^{11}$ OBs/g). To prepare the experimental virus suspension, HearNPV powder was dissolved in sterile water to a concentration of $1 \times 10^7$ OBs/ml, with the addition of 1% edible green dye (MedChemExpress) for visual confirmation of larval consumption. Each newly molted 4th instar larva was fed 2 µl of the virus suspension within 10 min (sterile water was used for the control group), and it was thereafter referred to as "infected larvae". This specific concentration and dosage of the virus consistently resulted in a lethal infection rate exceeding 95% in treated fourth-instar *H. armigera* larvae [26].

### Nile red staining

The fat body was dissected from healthy and infected larvae in PBS, fixed in 4% paraformaldehyde for 30 min at 25°C, and washed twice with PBS. Incubate the fat body in the dark with Nile Red (0.1 mg/mL, Beijing Coolaber Technology Co., Ltd, Beijing, China) and DAPI (0.05 mg/mL, Coolaber) for 15 min at 25°C, then wash three times with PBS. The samples were imaged using Leica SP8 confocal microscope (Leica Microsystems). Measure the area of each lipid droplet (LD) using ImageJ, and count a total of no less than 200 lipid droplets.

### Measurement of TAG and free fatty acids (FFA) levels

TAG levels were measured using a Triglyceride assay kit (Nanjing Jiancheng Bioengineering institute, Nanjing, China) and FFA levels were measured using a Nonesterified free fatty acids assay kit (Nanjing Jiancheng Bioengineering institute) according to the manufacturer's instructions. Briefly, collect fat body from 30 larvae per treatment group and weigh them, then homogenize the sample in PBS (w: v = 1 g: 9 mL). Then centrifuge the homogenate at 8, 000 rpm for 10 min at 4°C, and use the supernatant for TAG and FFA content measurement. For TAG content measurement, add 2.5 µl of supernatant and 250 µl of Enzyme reagent into a 96-well plate, and incubate them for 10 min at 37°C. Spectrum absorbance (OD) at 500nm wavelength was measured using a multimode microplate reader (Molecular Devices, California, USA) and then the content of TAG was calculated. For FFA content measurement, mix 200 µl of supernatant, 500 µl of Reagent 2 buffer, 1, 000 µl of Reagent 3 copper reagent, and 4, 000 µl of Reagent 1 evenly, then extract for 2 min. Centrifuge the mixture at 3, 500 rpm for 10 min. Take 2, 000 µl extract solution of underlayer and 250 µl of Chromogenic agent and mix them sufficiently, then incubate them at room temperature for 2 min. Subsequently, transfer the mixture in cuvettes of 1cm light path, measure OD values at 440nm for chromogenic reaction using a UV-VIS spectrophotometer (MAPADA, Shanghai, China). Three independent biological replicates were analyzed for each treatment.

### Behavior assays

**Climbing assays.**   To assess the climbing behavior of the infected larvae, vertical transparent glass tubes (300 mm height × 50 mm diameter), with 2cm-width wire mesh fixed at both

ends were used [27]. Treated *H. armigera* larvae (n = 50 larvae per treatment) were individually placed into the bottom of the glass tube containing a piece of artificial food ($20 \times 10 \times 10$ mm), sealed at both ends with transparent film, covered at the bottom with black cloth, and illuminated with LED white light (c. 500 lux) at the top. The height climbed by the larvae was recorded every 12 hours until all larvae either liquefied or pupated.

**Crawling assays.** Horizontal crawling assays were conducted using a horizontal PVC white tube (100cm length ×3cm width × 2cm height) in the darkroom. Treated *H. armigera* larvae were placed individually into one end of the tube, with an LED light (c. 500 lux) placed at the other end. The light was turned on simultaneously with timing, allowing the larvae to crawl horizontally for 5 min, and the straight-line distance crawled by the larvae was recorded. Each treatment group consisted of at least 50 larvae.

## Gene identification and sequence analysis

The sequences of *HaAKH1* (Genbank Accession: OP454989.1) and *HaAKH2* (Genbank Accession: OP454990.1) were identified from the central nervous system transcriptome of *H. armigera*, while the sequences of *HaAKHR* (LOC110384063), *HaLsd1* (LOC110380028), *HaLsd2* (LOC110380045), *HaBmm* (LOC110374396) and *HaHSL* (LOC110384566) were identified from the genomic sequence of *H. armigera* (Genbank Accession: GCF_030705265.1). Specific primers (S1 Table) were employed to amplify the full open reading frames of these genes, which were subsequently cloned into the pMD18-T vector (Takara Bio, Otsu, Japan) for sequencing (Ruibiotech, Beijing, China).

Homologous protein sequences of AKH and AKHR from different species were obtained from the GenBank database (S2 and S3 Tables). WebLogo v2.8.2 [28] was utilized to create sequence logos for AKH1 and AKH2. MEGA v6.0 [29] was used to build a neighbor-joining tree for AKHR with 1000 bootstrap replicates, where the Human Gonadotropin-Releasing Hormone Receptor (GnRHR, NP_000397.1) was designated as the outgroup.

## RT-qPCR

For mRNA expression analysis, total RNA was extracted using TRIzol reagent (Takara), and the quality and concentration of RNA were assessed using NanoDrop 2000 spectrophotometer. First-strand cDNA synthesis was performed using the PrimeScript II 1st Strand cDNA Synthesis Kit (Takara) according to the manufacturer's instructions. qPCR was conducted using SYBR Green Supermix (TaKaRa) on the CFX Connect TM Real-Time PCR System (Bio-Rad, CA, USA). In previous HearNPV studies, the *ribosomal protein L32* (*RPL32*) gene was identified as a stable house-keeping gene [30]. Before conducting gene expression analysis, the efficiencies of RT-qPCR primers were validated. The relative expression of each gene was analyzed by using $2^{-\Delta\Delta Ct}$ method [31] and the primers used in this study were listed in S1 Table.

## Western blotting

The total proteins of experimental samples were extracted using RIPA Lysis Buffer (Beyotime Biotechnology, Shanghai, China) with 1 mM PMSF. Protein concentrations were quantified by using BCA protein assay kit (Beyotime). Equal amounts of protein were separated by 12% SDS-PAGE and transferred to polyvinylidene fluoride membranes. The primary antibody against polyhedrin (poly) protein [27] (1:1000, Beijing Biosynthesis Biotechnology, Beijing, China) and the secondary antibody (goat anti-rabbit IgG conjugated with HRP, 1:10000, TransGen Biotech, Beijing, China) were used for poly protein. A mouse monoclonal antibody against β-actin (1:10000, TransGen) was used as a control. Immunoreactivity was imaged with the enhanced chemiluminescence using the Azure C600.

## dsRNA synthesis and RNAi bioassays

RNAi was performed by dsRNAs according to MEGAscript RNAi kit (Thermo Scientific, United states). Primers used for dsRNAs synthesis and the sizes of dsRNAs were listed in S1 Table. dsRNAs of *HaAKH1*, *HaAKH2*, *HaAKHR* and *HaHSL* were synthesized by the MEGAscript kit and then purified by the GeneJET RNA Purification kit (Thermo Scientific). The *EGFP* gene was used as control dsRNA (ds*EGFP*). All synthesized dsRNAs were diluted with DEPC water. The quality and quantity of dsRNAs were measured using the NanoDrop 2000 spectrophotometer.

The dsRNAs were injected into the proleg of each fourth-instar larva (10 μg per larva) within 24 h of moulting. To assess interference efficiency, brain or fat body samples of larvae (n ≥ 30 per treatment, three biological replicates) collected at 24 and 48 hours post-injection were subjected to RT-qPCR to measure the relative expression levels of the target genes. To determine the contents of TAG or FFA, and the mRNA and protein level of target genes, the fat body (n ≥ 30 per treatment, three biological replicates) was collected at 24 and 48 hours post-injection. The crawling assays were conducted at 48 hours post-injection (n ≥ 20 per treatment). For the climbing behavior and mortality rate statistical experiments, a second injection was administered 48 hours after the first injection, followed by climbing behavior experiments and mortality rate statistics (n ≥ 40 larvae per treatment).

## Mature peptides synthesis and injection bioassays

To predict the mature peptides, NeuroPred [32] was used for HaAKH1 and HaAKH2. The mature peptides of HaAKH1 and HaAKH2 were synthesized by Sangon Biotech (Shanghai, China), with a purity ≥ 95%. The mature peptides were diluted in PBS to 20 pmol and 2 μl was injected into the proleg of each fourth-instar larva within 24 h. Subsequently, at 24 and 48 hours post-injection, the fat body (n ≥ 30 per treatment, three biological replicates) was collected for TAG and FFA measurement, RT-qPCR and western blotting. The crawling assays were conducted (n ≥ 20 per treatment) at 48 hours post-injection. For the climbing behavior and mortality rate statistical experiments, a second injection was administered 48 hours after the first injection, followed by climbing behavior experiments and mortality rate statistics (n ≥ 40 larvae per treatment).

## Homology modeling and molecular docking

The three-dimensional structures of HaAKH1, HaAKH2 and HaAKHR were predicted by Phyre2 (http://www.sbg.bio.ic.ac.uk/phyre2/html/page.cgi?id=index). The molecular docking of HaAKH1 or HaAKH2 binding HaAKHR was performed by AlphaFold3 [33]. The docked models were visualized with PyMOL software (http://www.pymol.org/pymol).

## $Ca^{2+}$ imaging and fluorescence detection of Fluo-4 AM

The complete ORF sequence of HaAKHR was cloned into the pcDNA3.1(+)-mCherry vector to construct the recombinant vector of pcDNA3.1-mCherry-HaAKHR using the ClonExpress II One Step Cloning Kit (Vazyme, Nanjing, China) and then confirmed by sequencing (Ruibiotech). Endotoxin-free plasmid DNA of correct vector was extracted with EndoFree Plasmid Midi Kit (CWBio, Taizhou, China). HEK293T cells were used and transfected with this recombinant vector by Lipofectamine 2000 (Thermo Scientific). Fluo-4 Calcium Assay Kit (Beyotime) was used as the $Ca^{2+}$ fluorescent probe to detect intracellular $Ca^{2+}$ concentrations. Following the kit instructions, Fluo4-AM was co-incubated with transfected cells, and then

PBS or HaAKH1 or HaAKH2 was added to the cells. Changes in $Ca^{2+}$ signal fluorescence intensity were observed using Leica SP8 confocal microscopy.

## Subcellular localization

The complete ORF sequence of HaAKHR was cloned into the pEGFP-N1 vector to construct the recombinant vector of pEGFP-N1-HaAKHR using the ClonExpress II One Step Cloning Kit (Vazyme) and then confirmed by sequencing (Ruibiotech). Endotoxin-free plasmid DNA of correct recombinant vector and empty vector were extracted with EndoFree Plasmid Midi Kit (CWBio). HEK293T cells were used and transfected with this recombinant or empty vector by Lipofectamine 2000 (Thermo Scientific). Cell Plasma Membrane Staining Kit with DiD (Beyotime, Shanghai, China) was used to stain the transfected cell membranes. Leica SP8 confocal microscopy was used to observe the subcellular localization of HaAKHR.

## Elisa for HaHSL activity assay

To detect the enzyme activity of HsHSL, fat bodies from 30 larvae per treatment group were collected and weighed, then homogenized in PBS (w: v = 1 g: 9 mL). Centrifuge the mixture at 8, 000 rpm for 10 min, and use the supernatant for HaHSL activity assay. The Elisa of Insect HSL kit (Meike Biotechnology Co., Ltd, Suzhou, China) was used for HaHSL activity assay according to manufacturer's instructions. In brief, add 40 μl of Sample dilution and 10 μl of testing supernatant in Microelisa stripplate (coated with HSL antibodies) and incubate them for 30 min at 37˚C. Then add 50 μl HRP-Conjugate reagent to every well and incubate the plate for 30 min at 37˚C. For color rendering, add 50 μl of Chromogen solution A and 50 μl of Chromogen solution B to every well, and evade the light preservation for 10 min at 37˚C. Then add 50 μl of Stop solution to stop the reaction. Spectrum absorbance (OD) at 450nm wavelength was measured using a multimode microplate reader (Molecular Devices, California, USA).

## Data analysis

All statistical analyses were performed in GraphPad Prism 9.00 (GraphPad software, CA, USA). A single-sample Kolmogorov-Smirnov test was used to confirm that the data were normally distributed. Survival data were analyzed using the Mantel-Cox test followed by a log-rank test. Two-tailed Student's *t-tests* were used for two groups of data statistical analysis (*$p < 0.05$, **$p < 0.01$, ***$p < 0.001$), while one-way ANOVA, followed by a Tukey's HSD multiple comparison was used to analyze more than two groups ($p < 0.05$).

## Results

### HearNPV promoted host lipid metabolism and high-fat-diet induced climbing behavior

After infection with HearNPV, the body of *H. armigera* larvae became soft (S1A Fig) and the fat bodies became fragmented (S1B Fig). The sizes of LDs in fat body of infected larvae were significantly smaller than those of healthy larvae (Fig 1A and 1B). The TAG level in fat body of infected larvae was significantly decreased compared with healthy larvae (Fig 1C), and the level of free fatty acids (FFA), which was generated by hydrolysis of TAG, was significantly increased after infection (Fig 1D), suggesting enhanced lipid metabolism in the fat body after HearNPV infection.

To verify the relationship between lipid metabolism and HearNPV infection process, we added 10% coconut oil in the diet of *H.armigera* to make high-fat-diet (HFD) and fed it to the

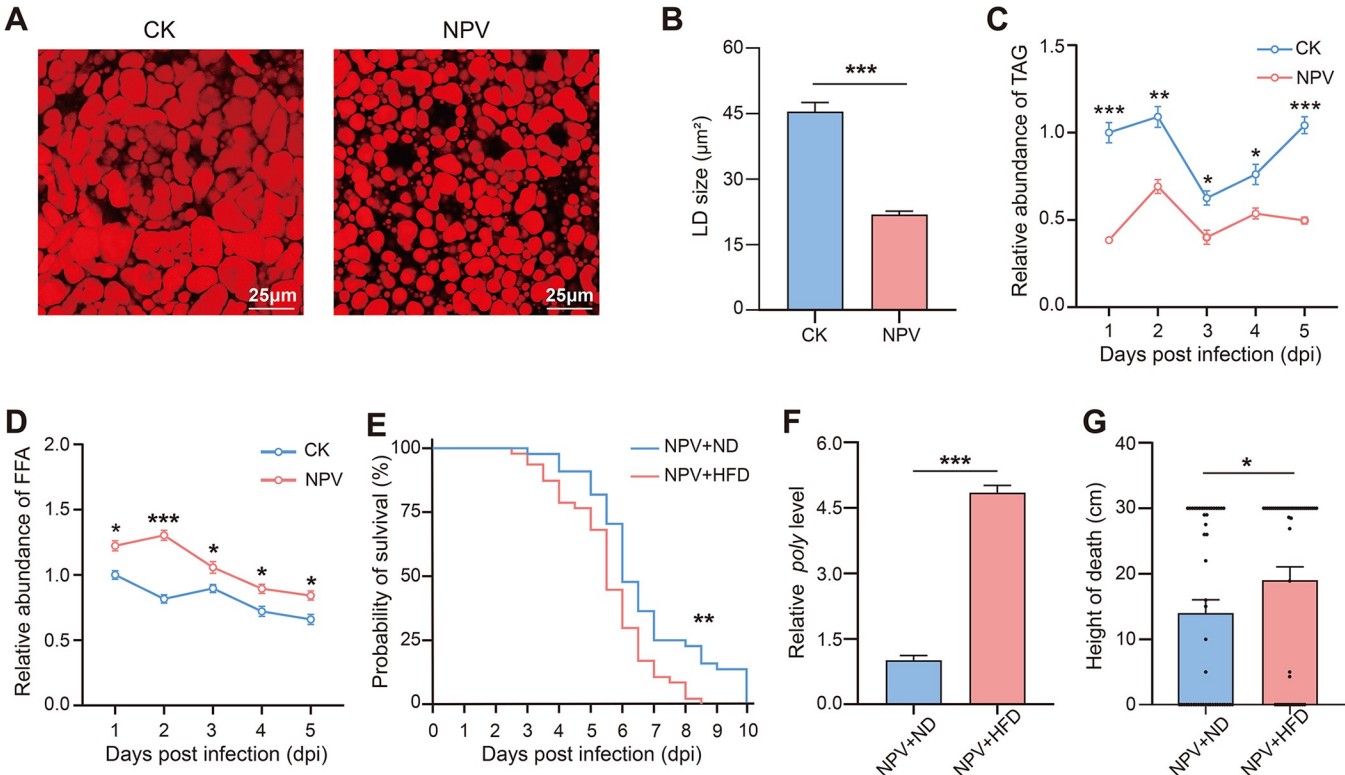

**Fig 1. Effects of HearNPV on lipid metabolism of *H. armigera* and effects of HFD on HearNPV-infected *H. armigera* larvae.** (A) LDs of healthy (CK) and HearNPV-infected (NPV) larvae at 2 dpi with Nile red. Scale bar: 25 μm. (B) LDs sizes of CK and NPV larvae. (C-D) TAG and FFA levels of CK and NPV larvae of *H. armigera* at 1, 2, 3, 4 and 5 days post-infection (dpi). (E) Survival of infected larvae with ND (NPV+ND) and infected larvae with HFD (NPV+HFD). (F) Relative expression level of *poly* in NPV+ND and NPV+HFD larvae. (G) Height at death of NPV+ND and NPV+HFD larvae. (Data were represented as mean ± SEM. *$p < 0.05$; **$p < 0.01$; ***$p < 0.001$).

infected larvae. Compared with the infected larvae fed with normal-diet (ND), the larvae fed with HFD had a faster mortality rate (Fig 1E) and a higher HearNPV replication (Fig 1F). We then explored the relationship between lipid metabolism and climbing behavior. After infection with HearNPV, the daily positions of HFD larvae were higher than those of ND larvae (S2 Fig). In addition, the infected HFD larvae had a significantly higher death height than those of ND larvae (Fig 1G), indicating that lipid metabolism played a considerable role in the HearNPV induced climbing behavior.

### *HaAKH1* and *HaAKH2* regulated lipid metabolism in infected larvae

The primary structures of HaAKH1 and HaAKH2 precursors were presented in S3A Fig. A phylogenetic analysis and a mature peptides analysis of AKHs were conducted and showed that HaAKH1 and HaAKH2 were highly conserved (S3B Fig and S2 Table). Tissue expression profiles of *HaAKH1* (S4A Fig) and *HaAKH2* (S4B Fig) showed that both *HaAKH1* and *HaAKH2* were expressed highly in the brain (including CC-CA). Compared with healthy larvae, the expression levels of *HaAKH1* (Fig 2A) and *HaAKH2* (Fig 2B) in infected larvae were significantly increased at 1, 2, 3, 4 and 5 days post-infection (dpi). Following the successful knockdown of *HaAKH1* and *HaAKH2* in the infected larvae (Figs 2C and 2D), the LDs size was significantly increased (Fig 2E and 2F), the TAG in the fat body was significantly accumulated (Fig 2G), and the FFA content was significantly decreased (Fig 2H), indicating that *HaAKH1* and *HaAKH2* promoted lipid metabolism in infected larvae. To further confirm this

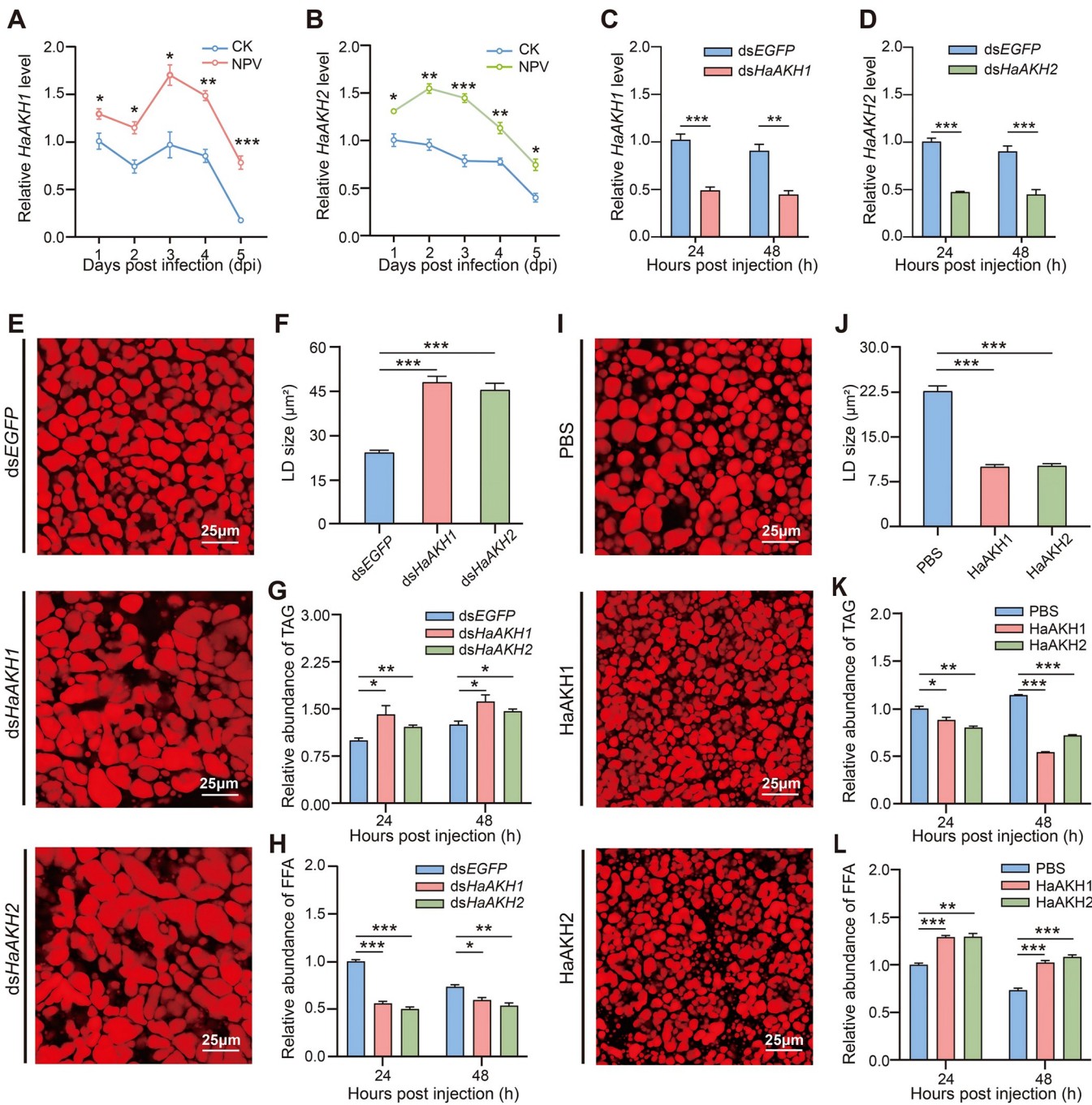

**Fig 2. *HaAKH1* and *HaAKH2* participated in lipid metabolism of HearNPV infected *H. armigera* larvae.** (A-B) Relative expression levels of *HaAKH1* and *HaAKH2* in healthy larvae (CK) and infected larvae (NPV). (C-D) Efficiency of RNAi of *HaAKH1* and *HaAKH2* in infected larvae treated with ds*HaAKH1* and ds*HaAKH2* for 24h and 48 h. (E) LDs of infected larvae treated with ds*EGFP*, ds*HaAKH1* and ds*HaAKH2* for 48 h. Scale bar: 25 μm. (F) Statistics of LDs sizes of infected larvae treated with ds*EGFP*, ds*HaAKH1* and ds*HaAKH2*. (G-H) TAG and FFA levels in infected larvae of *H. armigera* treated with ds*EGFP*, ds*HaAKH1* and ds*HaAKH2* for 24h and 48 h. (I) LDs of infected larvae treated with PBS, mature peptides of HaAKH1 and HaAKH2 for 48 h. Scale bar: 25 μm. (J) Statistics of LDs sizes of infected larvae treated with PBS, HaAKH1 and HaAKH2. (K-L) TAG and FFA levels of infected larvae of *H. armigera* treated with PBS and HaAKH1 and HaAKH2 for 24h and 48 h. (Data were represented as mean ± SEM. *$p < 0.05$; **$p < 0.01$; ***$p < 0.001$).

conclusion, HaAKH1 and HaAKH2 mature peptides were injected into infected larvae, which resulted in a significant reduction in LDs (Fig 2I and 2J), a significant decrease in TAG content (Fig 2K), and an increase in FFA content (Fig 2L).

### HaAKH1 and HaAKH2 induced climbing behavior of infected *H. armigera*

To determine the roles of the *HaAKH1* and *HaAKH2* in climbing behavior, we examined the changes in mortality rate, viral replication, and behaviors of larvae by RNAi of these genes and the injection of their mature peptides into the larvae. Following the successful knockdown of *HaAKH1* and *HaAKH2* in the infected larvae, the mortality rates of infected larvae were significantly reduced (Fig 3A). The amount of HearNPV was significantly decreased at both mRNA and protein levels (Fig 3B). In addition, the knockdown of *HaAKH1* and *HaAKH2* led to a decrease in the locomotion of both healthy and infected larvae (Fig 3C). Furthermore, knockdown of *HaAKH1* and *HaAKH2* resulted in a significant reduction in the height climbed before death (Fig 3D). Subsequently, we further confirmed our conclusion by injection of HaAKH1 and HaAKH2 active peptides. After treatment, the mortality rates of infected larvae were significantly accelerated (Fig 3E), and the virus replication was significantly increased (Fig 3F). The crawling distance (Fig 3G) and the death height (Fig 3H) were significantly increased compared with the control group.

### *In silico* and *in vitro* studies validated the HaAKH1-HaAKHR and HaAKH2-HaAKHR interaction

The sequence of *HaAKHR* was identified from the genome of *H. armigera*, and the cDNA sequence was cloned. The phylogenetic analysis was conducted to evaluate the association of HaAKHR with other AKHRs of other insects (S4 Fig and S3 Table). The HaAKHR was a typical Class A GPCR with seven transmembrane domains (S5 Fig). We constructed the ORF of HaAKHR into a subcellular localization expression vector and transfected it into HEK293T cells and found that HaAKHR was localized on the cell membrane (Fig 4A).

To explore whether the HaAKH1 and HaAKH2 interacted with HaAKHR, we conducted molecular docking of HaAKH1-HaAKHR (Fig 4B) and HaAKH2-HaAKHR (Fig 4C). The docking simulation prediction showed that HaAKH1 and HaAKHR bound through the formation of seven hydrogen bonds, and the six residues of HaAKHR (H37, Y186, Q188, S191, Q297 and K298) were crucial in the binding of HaAKH1. Similarly, HaAKH2 and HaAKHR formed eleven hydrogen bonds, involving nine key amino acid residues (D30, H37, R119, Y186, Q188, S191, Y275, Q297 and K298). In addition, fluorescence detection of Fluo-4 AM and $Ca^{2+}$ imaging were performed to determine whether HaAKH1 and HaAKH2 could increase intracellular $Ca^{2+}$ in cells expressing HaAKHR. As shown in Fig 4D and 4E, adding HaAKH1 and HaAKH2 to cells expressing HaAKHR significantly enhanced intracellular $Ca^{2+}$ signaling, indicating that HaAKH1 and HaAKH2 can strongly activate HaAKHR.

### HaAKHR regulated lipid metabolism

Tissue expression profile of *HaAKHR* showed that *HaAKHR* highly expressed in the fat body (S7 Fig). *HaAKHR* in infected larvae was significantly upregulated at 1, 2, 3, 4 and 5 dpi compared to healthy larvae (Fig 5A). After knockdown of *HaAKHR* by RNAi successfully (Fig 5B), the LDs size significantly increased (Fig 5C and 5D), the TAG content in the fat body significantly increased (Fig 5E), and the FFA content significantly decreased (Fig 5F). In addition, the mortality rate of infected ds*HaAKHR*-treated larvae was significantly reduced compared with the ds*EGFP* group (Fig 5G), and the HearNPV replication was significantly decreased at both the mRNA and protein levels (Fig 5H). The crawling distance (Fig 5I) of healthy and

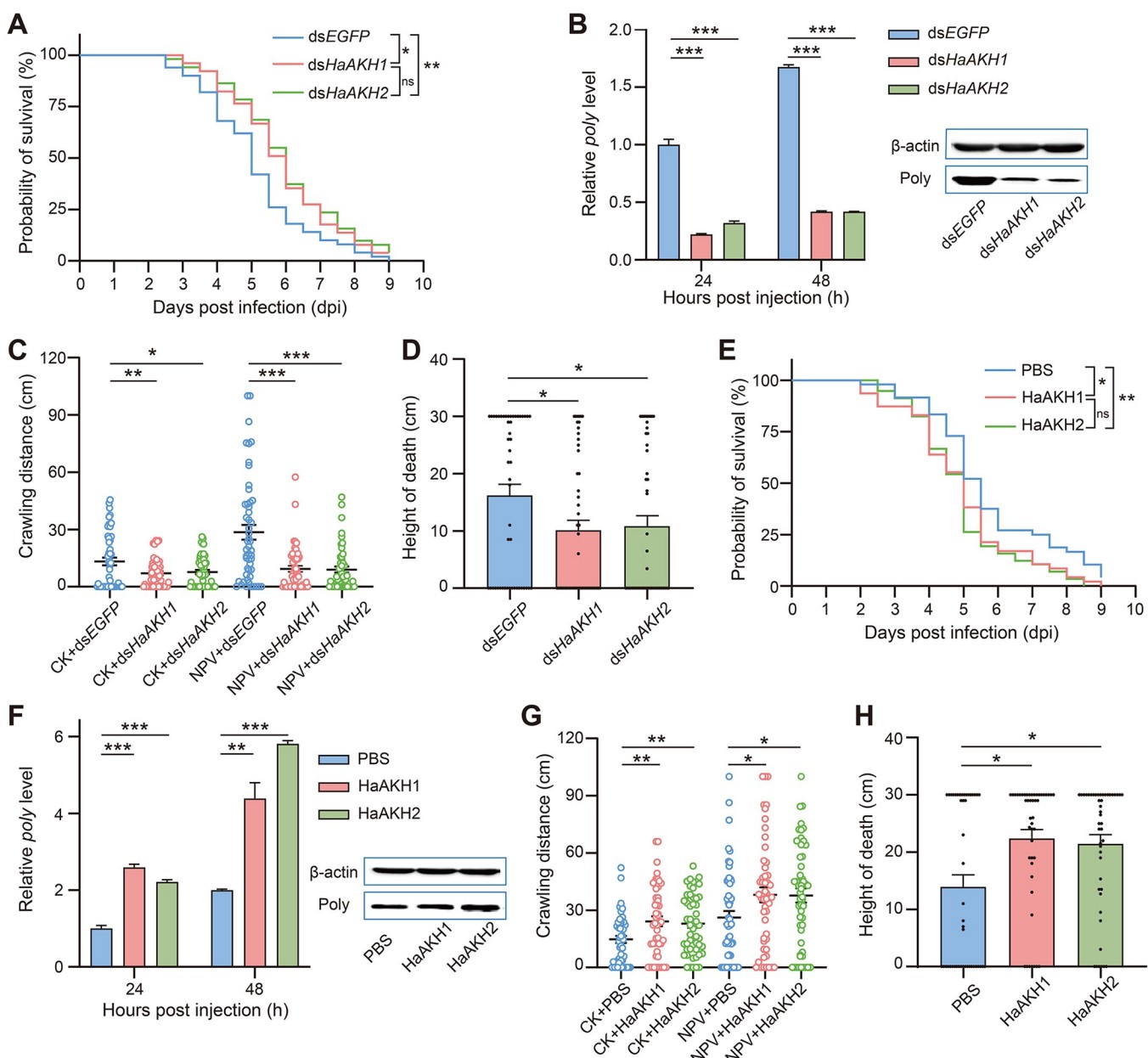

**Fig 3. *HaAKH1* and *HaAKH2* were involved in regulating the HearNPV infection process and behavior changes of *H. armigera*.** (A) Survival of infected larvae treated with dsRNAs (ds*EGFP*, ds*HaAKH1* and ds*HaAKH2*). (B) Effect of dsRNAs treatments on *poly* mRNA (24 and 48 hours post injection) and protein (48 hours post injection) expression level. (C) Crawling distance of healthy larvae treated with ds*EGFP* (CK+ds*EGFP*), ds*HaAKH1* (CK+ds*HaAKH1*) and ds*HaAKH2* (CK+ds*HaAKH2*) and infected larvae treated with ds*EGFP* (NPV+ds*EGFP*), ds*HaAKH1* (NPV+ds*HaAKH1*) and ds*HaAKH2* (NPV +ds*HaAKH2*). (D) Height at death of infected larvae treated with dsRNAs. (E) Survival of infected larvae treated with PBS and mature peptides of HaAKH1 and HaAKH2. (F) Effect of HaAKH1 and HaAKH2 treatments on *poly* mRNA (24 and 48 hours post injection) and protein (48 hours post injection) expression level. (G) Crawling distance of healthy and infected larvae treated with PBS, HaAKH1 and HaAKH2. (H) Height at death of infected larvae treated with PBS, HaAKH1 and HaAKH2. (Data were represented as mean ± SEM. *$p < 0.05$; **$p < 0.01$; ***$p < 0.001$).

infected larvae and the death height of infected larvae (Fig 5J) were both significantly decreased. Based on the above results, *HaAKHR* had the same effect as *HaAKH1* and *HaAKH2*, accelerating lipid metabolism in infected larvae, facilitating virus replication, and promoting locomotion and climbing behavior of infected larvae.

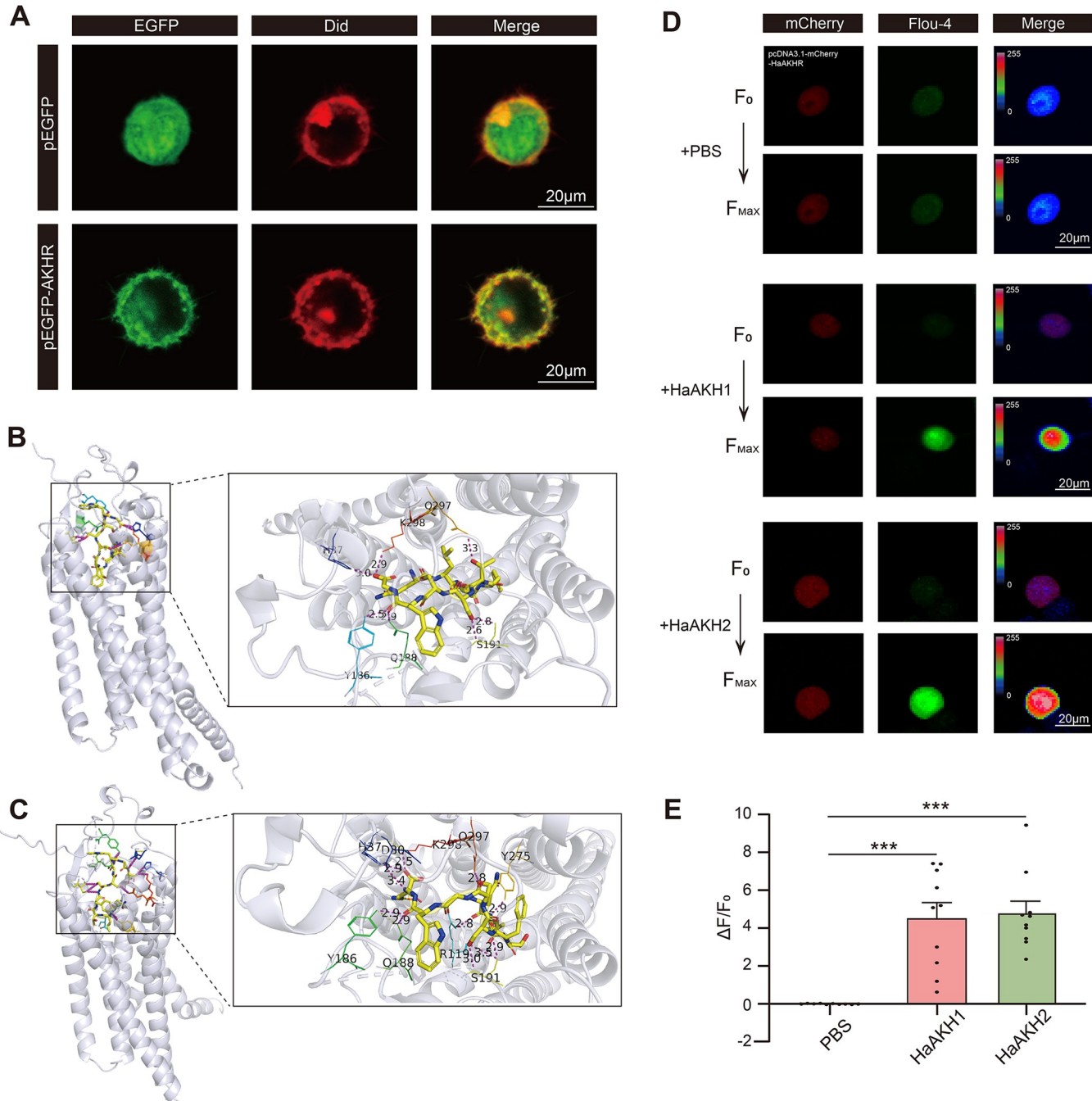

**Fig 4. *In silico* and *in vitro* studies validated the HaAKH1–HaAKHR and HaAKH2–HaAKHR interaction.** (A) Subcellular localization of HaAKHR proteins. pEGFP meant empty vector, pEGFP—AKHR meant vector expressing HaAKHR. EGFP: EGFP fluorescent signal; Did: Did fluorescent signal, labelling cell membrane; Merge: overlapping of EGFP and Did images. Scale bar: 20 μm. (B-C) Left: Structural overview of HaAKH1-HaAKHR and HaAKH2-HaAKHR complex models. The HaAKHR protein was shown in a cartoon representation in gray, and HaAKH1 and HaAKH2 were shown in a stick representation in yellow. The active-site residues were shown in the technical representation in rainbow. Right: Zoom-in view of the predicted interface. Key interface residues in HaAKHR were shown in the technical representation and were labeled by residue name and position. Hydrogen bonds were displayed in purple and labeled with their bond lengths (Å). (D) Representative images of $Ca^{2+}$ imaging after heterologous expression of HaAKHR in HEK293T cells in response to HaAKH1 and HaAKH2. pcDNA3.1—mCherry—HaAKHR meant the cells expressing the recombinant plasmid of HaAKHR with pcDNA3.1(+)—mCherry before treatment. $F_0$ represented the initial state and $F_{Max}$ represented the strongest state of $Ca^{2+}$ signal after treatment with PBS, HaAKH1 or HaAKH2. mCherry and Fluo-4 signal were shown in red and green, respectively; Merge: overlapping of mCherry and Fluo-4 images and shown in rainbow. Scale bar: 20 μm. (E) Fluorescence detection of Fluo-4 AM after heterologous expression of HaAKHR in HEK293T cells in response to PBS, HaAKH1 and HaAKH2. (Data were represented as mean ± SEM. *$p < 0.05$; **$p < 0.01$; ***$p < 0.001$).

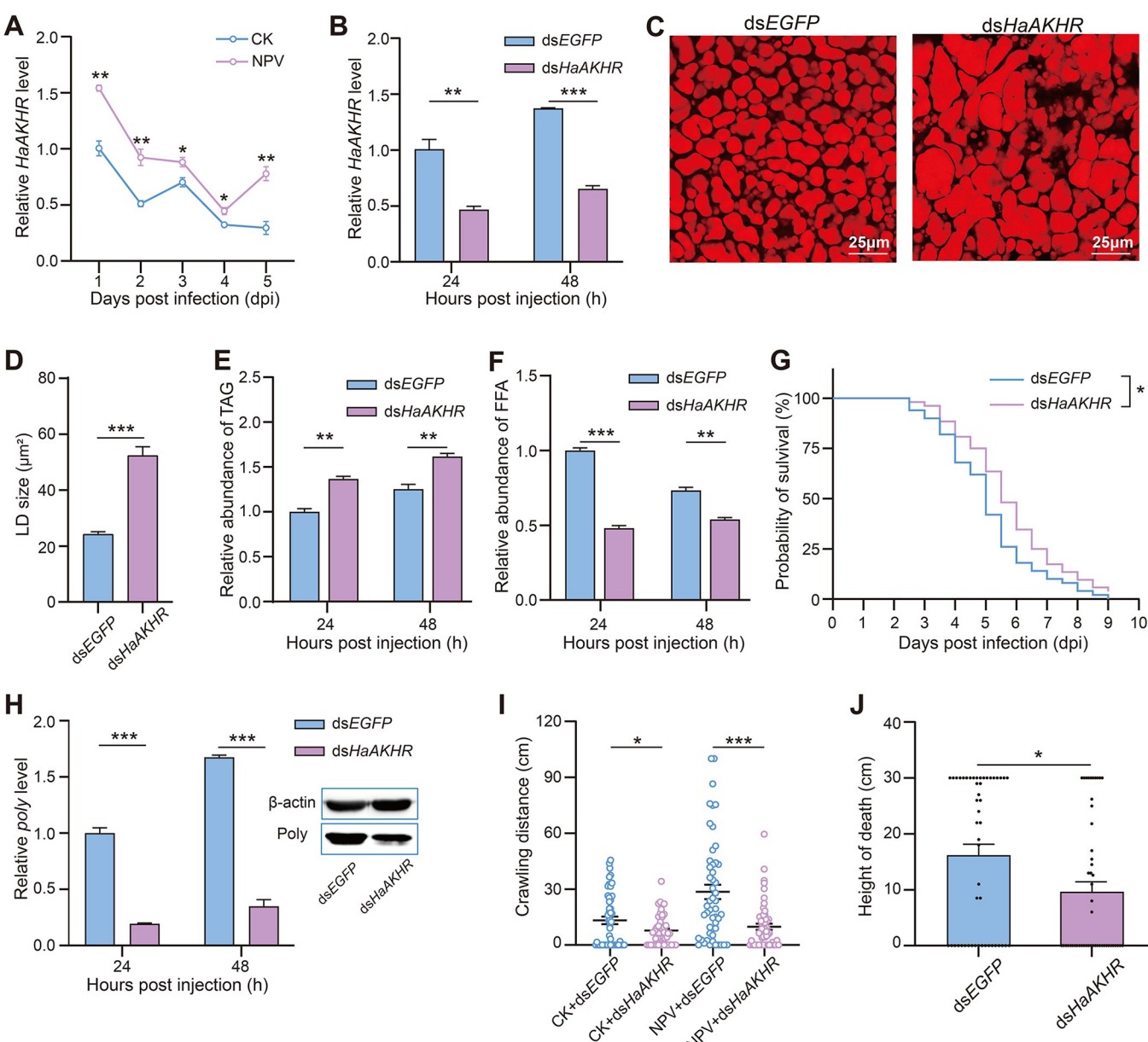

**Fig 5. *HaAKHR* involved in regulating the lipid metabolism, HearNPV infection process and behavioral changes of infected H. armigera.** (A) Relative expression level of *HaAKHR* in healthy (CK) and infected (NPV) larvae. (B) Efficiency of RNAi of *HaAKHR* in infected larvae treated with ds*HaAKHR* for 24h and 48 h. (C) LDs in fat bodies of infected larvae treated with ds*EGFP* and ds*HaAKHR* for 48 h. Scale bar: 25 μm. (D) Statistics of LDs sizes of infected larvae treated with ds*EGFP* and ds*HaAKHR*. (E-F) TAG and FFA levels of infected larvae of *H. armigera* treated with ds*EGFP* and ds*HaAKHR* for 24h and 48 h. (G) Survival of infected larvae treated with ds*EGFP* and ds*HaAKHR*. (H) Effect of ds*HaAKHR* on *poly* mRNA (24 and 48 hours post injection) and protein (48 hours post injection) expression level. (I) Crawling distance of healthy and infected larvae treated with ds*EGFP* and ds*HaAKHR*. (J) Height at death of infected larvae treated with ds*EGFP* and ds*HaAKHR*. (Data were represented as mean ± SEM. *$p < 0.05$; **$p < 0.01$; ***$p < 0.001$).

## Lipase HaHSL was regulated by HaAKH signaling

To investigate the downstream genes of HaAKH signaling, we identified four lipases, *H. armigera* hormone-sensitive triglyceride lipase (*HaHSL*), *H. armigera* Brummer (*HaBmm*), *H. armigera* Lipid storage droplet 1 (*HaLsd1*) and *H. armigera* Lipid storage droplet 2 (*HaLsd2*) by genome analysis. We detected the mRNA expression levels of these four triglycerideases in healthy and infected larvae at 1, 2, 3 and 4 dpi, and only *HaHSL* was significantly upregulated

at all four time points (Fig 6A), which was consistent with the expression patterns of *HaAKH* and *HaAKHR*. Subsequently, when *HaAKH1*, *HaAKH2* and *HaAKHR* were knocked down, the expression of *HaHSL* was also downregulated (Fig 6B), and after injection of HaAKH1 and HaAKH2, the expression of *HaHSL* was significantly upregulated (Fig 6C). In addition, the change of enzyme activity of HaHSL was also consistent with mRNA expression of *HaHSL* (Fig 6D–6F).

After knockdown of *HaHSL* (Fig 6G), the size of LDs (Fig 6H and 6I) and the TAG content (Fig 6J) were significantly increased, and the FFA content decreased (Fig 6K). The mortality rate of infected ds*HaHSL*-treated larvae was significantly reduced compared with the ds*EGFP* group (Fig 6L). In addition, the crawling distance of ds*HaHSL*-treated larvae was decreased compared with that of the ds*EGFP* group whether healthy or infected larvae (Fig 6M), and the death height was also significantly decreased (Fig 6N). Taken together, *HaHSL* was regulated by HaAKH signaling and HaHSL directly accelerated lipid metabolism, promoted virus replication, enhanced the locomotion of larvae, and induced climbing behavior of infected larvae.

## Discussion

Host lipids are the most important energy storage substances, which can provide energy and essential fatty acids for the survival of parasites when there is an energy demand [34,35]. In order to complete their life cycle, parasites must utilize the energy supply of their hosts, especially viruses, which need to fully utilize the energy metabolism of hosts for efficient replication and proliferation [36]. In this study, we found that the lipid homeostasis of the HearNPV infected larvae was greatly impaired, mainly manifested as enhanced TAG decomposition in fat body. The change in host lipid homeostasis manipulated by parasites is a broad event and has been reported in other organisms [14,37,38]. For example, *Rickettsia conorii*, a Gram-negative cytosolic intracellular bacterium, involves the host lipid droplet alterations [39]. After being parasitized by *Leptopilina boulardi*, the lipid content in the hemolymph of the host *D. melanogaster* larvae significantly decreased [40]. Therefore, exploring the molecular mechanisms behind this phenomenon may have universal significance. To explore the significance of HearNPV altering host lipid metabolism, we supplied HFD to infected larvae and found that HFD can greatly accelerate the mortality rate of infected larvae and the replication of virus, and significantly increase the death height of infected larvae. Based on this, we proposed that by manipulating the host lipid metabolism, HearNPV may promote its own replication and induce the host climbing behavior to facilitate its transmission.

The neuropeptide AKH-mediated lipolytic system is crucial for lipid supply and nutrient transfer [41], and the function of AKH is similar to that of glucagon in mammals in regulating lipid mobilization and carbohydrate levels. A previous study has described unusually high concentrations of glucagon in the plasma of patients with various bacterial infections [42]. Therefore, we speculated that the HearNPV may manipulate host lipid homeostasis by the AKH signaling, and therefore we proved this hypothesis by experiment. Although there is genome data of *H.armigera*, the *HaAKH* genes have not been annotated thoroughly. In this study, we identified two *HaAKH* genes, *HaAKH1* and *HaAKH2*, using our transcriptomic data, which was the first report of the existence of two *HaAKH* genes of *H.armigera* and was consistent with other Lepidoptera insects [43,44]. Subsequently, we found that both *HaAKH1* and *HaAKH2* were significantly upregulated after infection and proved that HaAKH signaling promoted lipid metabolism in infected larvae. Infection with *Plasmodium falciparum* activates AKH signaling and lipid mobilization in *Anopheles gambiae* [45]. Based on the above researches, it is a general phenomenon that parasites hijack AKH signaling to regulate host lipid metabolism.

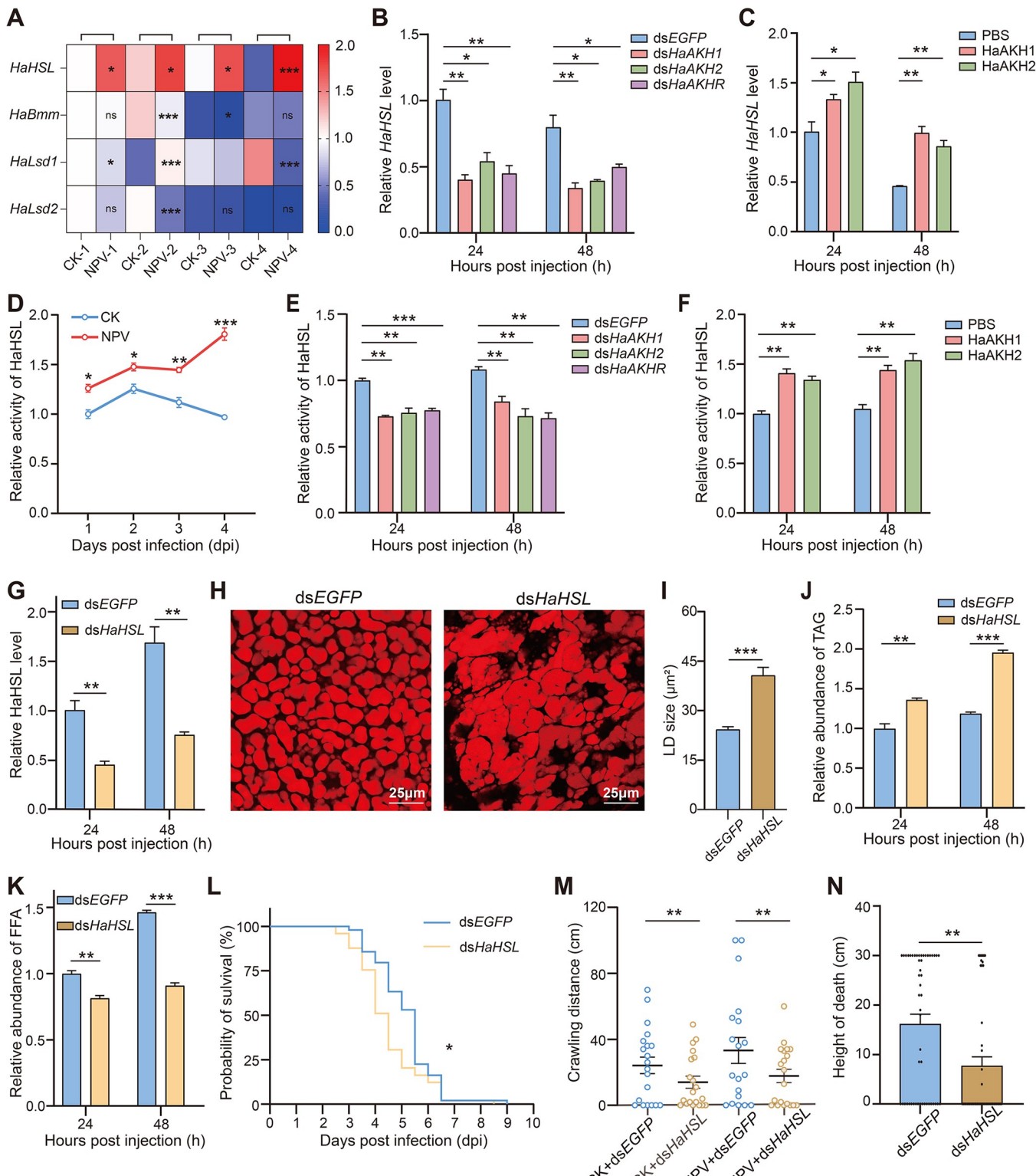

**Fig 6. *HaHSL* was regulated by *HaAKH/HaAKHR* and participated in the regulation of lipid metabolism and behavioral changes in infected *H. armigera* larvae.** (A) Relative expression level of four triglyceride enzymes (*HaHSL*, *HaBmm*, *HaLsd1* and *HaLsd2*) in healthy (CK) and infected (NPV) larvae at 1, 2,3 and 4 dpi. (B-C) Relative expression level of *HaHSL* in infected larvae of *H. armigera* treated with ds*EGFP*, ds*HaAKH1*, ds*HaAKH2* and ds*HaAKHR* or treated with PBS, HaAKH1 and HaAKH2 for 24h and 48 h. (D) Relative activity level of HaHSL in CK and NPV larvae. (E-F) Relative activity level of HaHSL of infected larvae treated with ds*EGFP*, ds*HaAKH1*, ds*HaAKH2* and ds*HaAKHR* or treated with PBS, HaAKH1 and HaAKH2 for 24 and 48 h. (G) Efficiency of

RNAi of *HaHSL* in infected larvae treated with ds*HaHSL* for 24h and 48 h. (H) LDs of infected larvae treated with ds*EGFP* and ds*HaHSL* for 48 h. Scale bar: 25 μm. (I) Statistics of LDs sizes of infected larvae treated with ds*EGFP* and ds*HaHSL*. (J-K) TAG and FFA levels in infected larvae of *H. armigera* treated with ds*EGFP* and ds*HaHSL* for 24h and 48 h. (L) Survival of infected larvae treated with ds*EGFP* and ds*HaHSL*. (M) Crawling distance of healthy larvae treated with ds*EGFP* (CK+ds*EGFP*) and ds*HaHSL* (CK+ds*HaHSL*), and infected larvae treated with ds*EGFP* (NPV+ds*EGFP*) and ds*HaHSL* (NPV+ds*HaHSL*). (N) Height at death of infected larvae treated with ds*EGFP* and ds*HaHSL*. (Data were represented as mean ± SEM. *$p < 0.05$; **$p < 0.01$; ***$p < 0.001$).

The function of AKH is mediated through the receptor AKHR [18–20], and we found that the characteristics of HaAKHR conformed to typical GPCRs features [41]. Through subcellular localization experiments, it was confirmed that HaAKHR was located on the cell membrane, which has also been proven in *B. mori* [44]. After being activated by ligands, GPCRs regulate the activity of related enzymes by coupling with G protein, resulting in the accumulation of second messenger cAMP, $Ca^{2+}$ mobilization, and ERK1/2 phosphorylation inside the cell. The activation effect of AKH on AKHR has been reported in insects such as *B. mori* [46] and *D. citri* [47], but there are no relevant reports in *H. armigera* yet. In addition, the identification of *HaAKH1*, *HaAKH2* and *HaAKHR* was predicted based on reports from other insects. Therefore, we used the molecular docking simulation prediction and cell heterologous expression system to confirm the activation of HaAKHR by HaAKH1 and HaAKH2 for the first time.

In this study, knockdown of *HaAKH1*, *HaAKH2* and *HaAKHR* of infected larvae significantly reduced the mortality rate and viral replication, and the mortality rate has been demonstrated mainly related to the amount of virus [48], which indicated that changes in host metabolism had a significant impact on HearNPV replication. Similarly, baculovirus LEF-11 hijacks the host ATPase family members and regulates the energy metabolism of the host *B. mori*, which efficiently promotes the multiplication of the virus [49]. Moreover, fatty acid synthase involved in energy metabolism is necessary for the effective replication of AcMNPV in *Spodoptera frugiperda* cells [50]. In addition, baculoviruses not only regulate host lipid metabolism, but also induce the climbing behavior, and the quantity of the baculovirus is highly correlated with climbing death height [27]. Therefore, HaAKH signaling promoted lipid metabolism, which caused an increase in viral replication, further leading to the climbing behavior of infected larvae. From another perspective, we found that HaAKH signaling also promoted the locomotion of infected larvae, which indicated that HearNPV utilized host HaAKH signaling to regulate the climbing behavior by improving locomotion ability of infected larvae. Furthermore, it has been confirmed that AKH signaling has a role in regulating locomotion [51].

Although the AKH signaling can regulate lipid metabolism, AKH and AKHR cannot directly break down the lipid. Therefore, we explored the downstream of HaAKH signaling which directly regulated lipid metabolism in infected larvae. Consistent with *D. melanogaster*, we also identified two HaLsds and two types of lipases promote lipolysis, HaBmm and HaHSL, in *H. armigera* [52,53]. In the present study, we found that the transcriptional expression level and enzyme activity of HaHSL were significantly upregulated after HearNPV infection, and changed with the HaAKH signaling system. Moreover, HaHSL also participated in lipid metabolism of infected larvae, indicating that HaHSL may be the downstream of HaAKHR. However, a previous report suggests that AKH signaling activates PKA, which phosphorylates Bmm and Lsd2, thus inducing lipolysis in fat body cells [23], which is inconsistent with the results of our present study. The reason for this difference is presumably to be differences between different species, or it is probably that the HaAKH signaling plays a different mechanism than normal under HearNPV infection, which requires more researches to explore this issue. In addition, HaHSL also participated in behavioral changes induced by HearNPV, further confirming the conclusion that HearNPV hijacked the HaAKH signaling to regulate lipid metabolism, thereby manipulating climbing behavior of infected larvae.

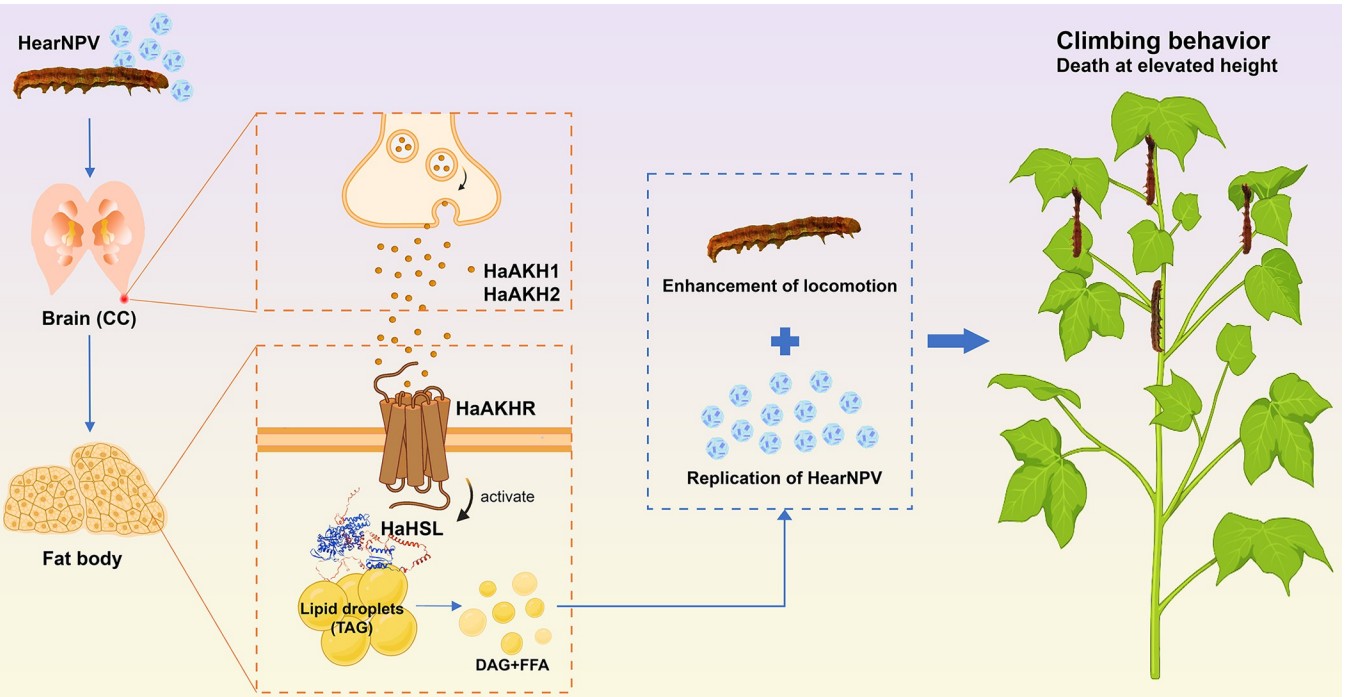

**Fig 7. Model for HearNPV manipulating host HaAKH signaling to induce the climbing behavior.** HearNPV led to an increase in the expression of *HaAKH1* and *HaAKH2*, and the secreted HaAKH1 and HaAKH2 peptides bound to the receptor HaAKHR located on the cell membrane of the fat body, inducing the lipase HaHSL to break down TAG in fat body, providing energy for virus replication and promoting the locomotion of infected larvae, ultimately inducing the climbing behavior. Image created with Biorender.com, with permission.

Most studies believe that climbing behavior induced by baculoviruses is more conducive to the spread of viruses. A research speculates that behavioral changes of *Mamestra brassicae* larvae induced by baculovirus most probably benefit the virus [54], while the climbing behavior favors horizontal transmission of *S. exigua* MNPVs (SeMNPVs) via intraspecific necrophagy in *S. exigua* [25,55]. In addition, the location where the cadavers are exposed at high altitude is more likely to be carried by wind, parasitic wasps, predators, and scavengers, who may spread the virus over longer distances [13,56,57]. Based on this, we speculated that the baculoviruses enhance their own replication by manipulating host lipid metabolism via AKH signaling, accelerating the use of host energy to promote its own replication, inducing host climbing behavior, and ultimately facilitating their own transmission. Clearly, further studies are required to clarify the ecological significance of climbing behavior mediated by AKH significantly.

In conclusion, we proposed a model in which baculoviruses manipulate host behavior by hijacking the AKH signaling, as shown in Fig 7. The baculovirus hijacked the host AKH signaling, and improved lipid metabolism through lipase HSL, which provided energy for virus replication and promoted the locomotion of infected larvae, which jointly promoted the climbing behavior of infected larvae. Overall, our research not only broadens the understanding of the molecular mechanisms by which parasites manipulate host behavior, but also provides valuable insights for further integrated pest management by targeting the AKH signaling pathway.

## Supporting information

**S1 Fig. The effect of HearNPV on the *H. armigera* larvae.** (A) Photos of the healthy (CK) and HearNPV-infected (NPV) 4th instar *H. armigera* larvae. The arrow indicated that the

body of infected larva is softer than the healthy larva. Scale bar: 5 mm. (B) Photos of fat bodies of CK and NPV larvae. Scale bar: 10 mm.
(TIF)

**S2 Fig. The effect of high-fat diet on the height of infected larvae.** Mean (±SEM) height of infected larvae with normal-diet (NPV+ND) and infected larvae with high-fat-diet (NPV +HFD). ***$p < 0.001$.
(TIF)

**S3 Fig. Sequence analysis of *Ha*AKH1 and *Ha*AKH2 precursors in *H.armigera*.** (A) Schematic diagram showing protein characteristics for *Ha*AKH1 and *Ha*AKH2. (B) Phylogenetic analysis of AKHs of insect species. *H.armigera* sequences were highlighted in red. (C) Alignment of sequence of AKH mature peptides in different species. The calculated consensus logo was shown at the bottom. The protein names and accession numbers were listed in S2 Table.
(TIF)

**S4 Fig. Spatial transcript expression analysis of *HaAKH1* and *HaAKH2* in *H.armigera* larvae.** (A) Relative expression level of *HaAKH1* in larvae various tissues. (B) Relative expression level of *HaAKH2* in larvae various tissues. Data represented mean ± SEM. Different lowercase letters indicated significant differences among different tissues ($p < 0.05$).
(TIF)

**S5 Fig. Phylogenetic relationship of AKHR from insects.** *H.armigera* sequences were highlighted in red. The protein names and accession numbers were listed in S3 Table.
(TIF)

**S6 Fig. Schematic diagram of HaAKHR.** Seven transmembrane domains were located within the orange shaded cell membrane. The conserved motifs of Rhodopsin-like receptors were highlighted in red. A pair of cysteine residues that form a disulfide bond were highlighted in blue. The predicted N-glycosylation sites were highlighted in green.
(TIF)

**S7 Fig. Spatial transcript expression analysis of *HaAKHR* in *H.armigera* larvae.** Relative expression level of *HaAKHR* in larvae various tissues. Data represented mean ± SEM. Different lowercase letters indicated significant differences among different tissues ($p < 0.05$).
(TIF)

**S1 Table. Primers used in this study.**
(DOCX)

**S2 Table. GenBank accession numbers used for the multiple sequence alignment and phylogenetic analysis of AKH and ACP.**
(DOCX)

**S3 Table. GenBank accession numbers used for the multiple sequence alignment and phylogenetic analysis of AKHR.**
(DOCX)

## Author Contributions

**Conceptualization:** Lin Zhu, Zhen Li, Xiaoxia Liu.

**Data curation:** Lin Zhu, Yuqing Xie, Xiaoxia Liu.

**Formal analysis:** Lin Zhu, Yuqing Xie, Chenxi Liu, Zhongjian Shen, Limei Cai, Xinyuan Ning.

**Funding acquisition:** Xiaoming Liu, Qiuying Huang.

**Investigation:** Lin Zhu, Yuqing Xie, Jie Cheng, Limei Cai, Xinyuan Ning.

**Methodology:** Lin Zhu, Chenxi Liu, Jie Cheng, Songdou Zhang, Xiaoxia Liu.

**Project administration:** Xiaoming Liu, Xiaoxia Liu.

**Resources:** Songdou Zhang.

**Software:** Lin Zhu, Limei Cai, Xinyuan Ning.

**Supervision:** Chenxi Liu, Zhen Li, Qiuying Huang, Xiaoxia Liu.

**Validation:** Lin Zhu, Yuqing Xie, Zhongjian Shen.

**Visualization:** Lin Zhu, Yuqing Xie.

**Writing – original draft:** Lin Zhu, Yuqing Xie, Jie Cheng, Zhongjian Shen, Songdou Zhang, Xiaoxia Liu.

**Writing – review & editing:** Lin Zhu, Chenxi Liu, Xiaoming Liu, Zhen Li, Xiaoxia Liu.

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
