## [Decision Letter · Decision Letter 0]

30 Dec 2024

PPATHOGENS-D-24-02518

Baculoviruses manipulate host lipid metabolism via adipokinetic hormone signaling to induce climbing behavior

PLOS Pathogens

Dear Dr. Zhu,

Thank you for submitting your manuscript to PLOS Pathogens. After careful consideration, we feel that it has merit but does not fully meet PLOS Pathogens's publication criteria as it currently stands. Therefore, we invite you to submit a revised version of the manuscript that addresses the points raised during the review process.

Please submit your revised manuscript within 30 days Feb 28 2025 11:59PM. If you will need more time than this to complete your revisions, please reply to this message or contact the journal office at plospathogens@plos.org. Please include the following items when submitting your revised manuscript:

We look forward to receiving your revised manuscript.

Kind regards,

Eain A Murphy, Ph.D.

Academic Editor

PLOS Pathogens

Robert Kalejta

Section Editor

PLOS Pathogens

Sumita Bhaduri-McIntosh

Editor-in-Chief

PLOS Pathogens

orcid.org/0000-0003-2946-9497

Michael Malim

Editor-in-Chief

PLOS Pathogens

orcid.org/0000-0002-7699-2064

**Additional Editor Comments:**

Dear Dr. Zhu,

PLoS Pathogens sent your manuscript to be reviewed by three experts in the lipidomics and/or Baculovirus field. All three reviewers were supportive of this work and suggested modifications to the text which will improve readability and clarity. As such, a decision of Minor Modify was rendered. Please address the concerns of the three reviewers and resubmit a modified version of the manuscript.

Sincerely,

Eain A. Murphy Ph.D.

**Journal Requirements:**

At this stage, the following Authors/Authors require contributions: Lin Zhu, Yuqing Xie, Jie Cheng, Zhongjian Shen, Xiaoming Liu, Limei Cai, Xinyuan Ning, Chenxi Liu, Songdou Zhang, Zhen Li, Qiuying Huang, and Xiaoxia Liu. Please ensure that the full contributions of each author are acknowledged in the "Add/Edit/Remove Authors" section of our submission form.

https://journals.plos.org/plospathogens/s/submission-guidelines#loc-parts-of-a-submission

Potential Copyright Issues:

- Please confirm that you are the photographer of Figure 1A, or provide written permission from the photographer to publish the photo(s) under our CC BY 4.0 license.

- Figure 7; Please confirm whether you drew the images / clip-art within the figure panels by hand. If you did not draw the images, please provide a link to the source of the images or icons and their license / terms of use; or written permission from the copyright holder to publish the images or icons under our CC BY 4.0 license. Alternatively, you may replace the images with open source alternatives. See these open source resources you may use to replace images / clip-art:

6) Please amend your detailed Financial Disclosure statement. This is published with the article. It must therefore be completed in full sentences and contain the exact wording you wish to be published. State the initials, alongside each funding source, of each author to receive each grant. For example: "This work was supported by the National Institutes of Health (####### to AM; ###### to CJ) and the National Science Foundation (###### to AM)." State what role the funders took in the study. If the funders had no role in your study, please state: "The funders had no role in study design, data collection and analysis, decision to publish, or preparation of the manuscript.".

**Reviewers' Comments:**

Reviewer's Responses to Questions

**Part I - Summary**

Reviewer #1: This is a detailed and very thorough analysis of the role of lipid metabolism in baculovirus-induced climbing behavious of infected larvae. The results provide new and interesting data to help explain why baculovirus induces climbing behaviour by manipulating energy sources, although the link to why climbing is impacted rather than just crawling remains elusive. Overall, this is of significance not just to those that study baculoviruses but more broadly as well.

Overall the study appears to have been carefully carried out with appropriate data and controls. However, the nature of the figures with many sub-panels means that the imaging data is quite hard to discern in the very small panels. Much is made of the difference in size observed for lipid droplets but sometimes this is hard to distinguish. The use of the pink and orange for colours is mostly suitable but in some line graphs e.g. Fig 3A, it is hard to see the difference between the orange and the pink. I wonder if one of the colours should be changed to something more distinctive?

The abstract is clear and concise except for the phrase in line 39/40 - this is ambiguous and as written could mean the height of the larvae rather than the height travelled or reached by the larvae.

The materials and methods section is clear and sufficiently detailed.

The results section is well presented - I picked up a few typogaphical errors, minor mistakes in English grammar that should be corrected e.g. 277 suggesting enhanced (the not needed); 304 post-infection; 303 & 305, 'in' infected larvae (not 'of'); 306, remove and after the comma; 332 (Figure S4 'and' Table S3).

I liked the use of the schematic to put forward the proposed mechamism of action.

279 - Why was coconut oil used to increase lipds in the diet?

319/320 - Reference for enhancement of crawl

Reviewer #2: In this study, the authors investigate how baculovirus influences lipid metabolism in infected larvae and the subsequent behavioral changes, particularly climbing behavior, which is believed to facilitate virus transmission. Using various techniques, the authors have demonstrated that manipulating lipid metabolism through HaAKH signaling not only impacts the survival and replication of the virus but also enhances the climbing behavior of the larvae. They have also shown that hormone-sensitive lipase activity is regulated by HaAKH signaling and is essential for the climbing behavior induced by the virus. The findings of this study offer insights into the molecular mechanisms by which baculoviruses manipulate host behavior through lipid metabolism. The authors propose a model in which baculoviruses hijack AKH signaling to enhance lipid metabolism, thereby promoting their own replication and inducing climbing behavior in infected larvae. This research expands the understanding of host-parasite interactions. Although the topic investigated is intriguing, there are several issues with writing, describing methodology procedures, and presenting results that require serious revision (possibly necessitating a revision and resubmission).

Reviewer #3: Climbing behavior is a renowned phenotype for caterpillars when they were infected by baculoviruses. However, the underlying molecular mechanisms are largely unclear. Zhu et al., performed the experiments to investigate how Helicoverpa armigera single nucleopolyhedrovirus (HearNPV) influences host behavior, specifically promoting climbing behavior in infected larvae through manipulation of lipid metabolism. They further linked host lipid metabolism to behavioral changes, focusing on the roles of two adipokinetic hormones (HaAKH1 and HaAKH2) and their receptor, HaAKHR. One of the key findings is that hormone-sensitive lipase (HaHSL) acts as a downstream effector of HaAKH signaling, bridging the gap between AKH receptor and changes of host lipid droplets. Their findings provided novel insights into how HearNPV affects its host's climbing behavior. Overall, I really enjoyed reading this manuscript and thought this study is interesting and well-done.

**Part II – Major Issues: Key Experiments Required for Acceptance**

Reviewer #1: I have no major issues with this paper.

Reviewer #2: L271: How were softness and shallowness measured? These aspects are not evident in the figure. The figure should be informative, and the differences should be highlighted using arrowheads, etc.

Figures 1, 2, 5, and 6: The indicated differences in the microscopy figures are unclear. The pointed differences should be clearly indicated. Perhaps microscopic images with higher magnifications would better facilitate the comparison of differences.

Reviewer #3: 1. In Figures 1, 2, 5, and 6, the authors compared the size of larval lipid droplets, but only showed some representative images. A more thorough quantitative comparison is necessary to strengthen the conclusions and the n value should be provided. Specifically, it would be helpful to include measurements of the lipid droplet size to better evaluate the observed changes, please also provide the detailed methods in the Method section.

2. The authors found that the crawling distance of dsHaHSL-treated healthy larvae was decreased compared with the dsEGFP-treated healthy larvae. How about the healthy larvae that dsHaAKH1-treated, dsHaAKH2-treated or dsHaAKHR-treated? It needs to be clear whether dsHaAKH or dsHaAKHR alone can cause changes in crawling ability.

**Part III – Minor Issues: Editorial and Data Presentation Modifications**

Reviewer #1: Minor issues and data presentation suggestions have been incorporated into the main review section.

Reviewer #2: Some parts of the introduction appear fragmented and require revision for better cohesion.

L70: triacylglycerol (TAG)

L87: interactions between parasites and hosts; for instance, injecting AKH into locusts infected with Metarhizium anisopliae accelerated the ...

L108: established from what? A natural population or a lab-reared one?

L110: larvae were reared ...

L112: moved or transferred?

L118: How were the larvae fed? Force-feeding or feeding via an artificial diet?

L123-129: Is it feasible to conduct standing and microscopy observations without preparing thin sections of the tissue (microtome)?

L130: free fatty acids (FFA)

L134-137: This section needs to be rewritten. It appears that the kit protocol has been copied without any modifications!

L145: the height climbed by the larvae was recorded...

L149: infected H. armigera larvae instead of treated H. armigera larvae

L176: qPCR instead of qRT-PCR; as cDNA is synthesized beforehand.

L180: RT-qPCR instead of qRT-PCR; as cDNA synthesis and qPCR were performed separately. Please correct this throughout the manuscript.

L188: poly?

L196: What are the sizes of the dsRNAs? How were they diluted? Please provide more details.

L216: ... and 2 µl was injected into the...

L235: Why did the authors use HEK293T cells instead of insect-specific cells like HZFB cells?

L252-258: This section needs to be rewritten. It seems that the kit protocol has been copied without any modifications!

L271: How were softness and shallowness measured? These aspects are not evident in the figure. The figure should be informative, and the differences should be highlighted using arrowheads, etc.

Figures 1, 2, 5, and 6: The indicated differences in the microscopy figures are unclear. The pointed differences should be clearly indicated. Perhaps microscopic images with higher magnifications would better facilitate the comparison of differences.

L289-299: This section is unnecessary and can be moved to supplementary texts.

L304: Following the successful knockdown of HaAKH1 and HaAKH2 in the infected larvae...

L316: RNAi of these genes and the injection of their mature peptide into the larvae.

L322: ...in the height climbed before death.

L326-327: This seems contradictory! These parameters should be increased, as shown in the figures.

L328-348: The interaction of HaAKH with HaAKHR is evident. Did the authors anticipate a different result?

L349: “HaAKHR regulated lipid metabolism” instead of the current title.

L418: It is unclear whether Candidatus Liberibacter asiaticus can be considered a parasite. It is a plant pathogen vectored by Diaphorina citri.

L426-436: This section is not particularly relevant to the discussion.

L437: This should be deleted.

Reviewer #3: 1. The authors report interference efficiency at 24 and 48 hours after injection, with two time points showing around 50% efficiency, indicating effective interference within 48 hours. However, in the Methods section (lines 208-211), the second injection is 48 hours after the first shot. Is this a mistake? In addition, the authors should clarify why this specific time point was chosen for the second injection.

2. In the Crawling assays, there is a discrepancy between the CK-dsEGFP group in Figure 3C and the CK+dsEGFP group in Figure 6L, both representing healthy larvae treated with dsEGFP. This difference is substantial and affects the interpretation of the study’s conclusions. The authors should address the reasons behind this discrepancy between the two controls.

3. There is inconsistency in the use of "hpi" and "dpi" across different figures, with some figure legends also reversing these terms. It is recommended to standardize the terminology and use "dpi" consistently throughout the manuscript.

4. In Figure 1E-G and Figure 6L, the notation "xx+xx" is used in the figures, while "xx-xx" is used in the figure legends. It is necessary to use the same format.

5. Figure S1 does not indicate statistical significance, which should be addressed.

6. The image of western blot does not indicate the hours post injection.

7. Line 379-381 describe the results of Figure 6L, not Figure 6K. There is no description of Figure 6K in Results.

8. In Line 415, the reference format differs from the rest of the manuscript. Please ensure that all references are consistently formatted.

PLOS authors have the option to publish the peer review history of their article (what does this mean?). If published, this will include your full peer review and any attached files.

Reviewer #1: No

Reviewer #2: No

Reviewer #3: **Yes: **Jianhua Huang

**Figure resubmission:**
---

## [Editor Report · Decision Letter 1]

23 Jan 2025

Dear Dr. Zhu,

We are pleased to inform you that your manuscript 'Baculoviruses manipulate host lipid metabolism via adipokinetic hormone signaling to induce climbing behavior' has been provisionally accepted for publication in PLOS Pathogens.

Best regards,

Eain A Murphy, Ph.D.

Academic Editor

PLOS Pathogens

Robert Kalejta

Section Editor

PLOS Pathogens

Sumita Bhaduri-McIntosh

Editor-in-Chief

PLOS Pathogens

orcid.org/0000-0003-2946-9497

Michael Malim

Editor-in-Chief

PLOS Pathogens

orcid.org/0000-0002-7699-2064

Dear Dr. Zhu,

your manuscript was reviewed by the editorial staff here at PLoS Pathogens and based on the significant modifications to the offering in light of the previous comments, we have come to a decision of Accept without additional external review. Congratulations. this is a nice piece of work and your team should be commended on a nice study.

Cheers,

Eain Murphy Ph.D.
---

## [Editor Report · Acceptance letter]

26 Jan 2025

Dear Dr. Zhu,

We are delighted to inform you that your manuscript, "Baculoviruses manipulate host lipid metabolism via adipokinetic hormone signaling to induce climbing behavior," has been formally accepted for publication in PLOS Pathogens.

Best regards,

Sumita Bhaduri-McIntosh

Editor-in-Chief

PLOS Pathogens

orcid.org/0000-0003-2946-9497

Michael Malim

Editor-in-Chief

PLOS Pathogens

orcid.org/0000-0002-7699-2064